# Changes in Serum Levels of Ketone Bodies and Human Chorionic Gonadotropin during Pregnancy in Relation to the Neonatal Body Shape: A Retrospective Analysis

**DOI:** 10.3390/nu14091971

**Published:** 2022-05-09

**Authors:** Kiwamu Noshiro, Takeshi Umazume, Rifumi Hattori, Soromon Kataoka, Takashi Yamada, Hidemichi Watari

**Affiliations:** 1Department of Obstetrics and Gynecology, Graduate School of Medicine, Hokkaido University, Sapporo 060-8638, Japan; ichigoichie_finalfantasy@hotmail.com (K.N.); watarih@med.hokudai.ac.jp (H.W.); 2Department of Obstetrics and Gynecology, Obihiro-Kosei General Hospital, Obihiro 080-0024, Japan; rrrrrr4756@yahoo.co.jp; 3Department of Obstetrics and Gynecology, Hakodate Central General Hospital, Hakodate 040-8585, Japan; sorokata@hakochu-hp.gr.jp; 4Department of Obstetrics and Gynecology, Japan Community Health Care Organization Hokkaido Hospital, Sapporo 062-8618, Japan; yamatakashi@me.com

**Keywords:** ketone body, pregnancy, morning sickness, hyperemesis gravidarum, human chorionic gonadotropin (HCG)

## Abstract

Among the physiological changes occurring during pregnancy, the benefits of morning sickness, which is likely mediated by human chorionic gonadotropin (HCG) and induces serum ketone production, are unclear. We investigated the relationship between serum levels of ketone bodies and HCG in the first, second, and third trimesters and neonatal body shape (i.e., birth weight, length, head circumference, and chest circumference) in 245 pregnant women. Serum levels of 3-hydroxybutyric acid peaked in late-stage compared with early stage pregnancy (27.8 [5.0–821] vs. 42.2 [5.0–1420] μmol/L, median [range], *p* < 0.001). However, serum levels of ketone bodies and HCG did not correlate with neonatal body shape. When weight loss during pregnancy was used as an index of morning sickness, a higher pre-pregnancy body mass index was associated with greater weight loss. This study is the first to show that serum ketone body levels are maximal in the third trimester of pregnancy. As the elevation of serum ketone bodies in the third trimester is a physiological change, high serum levels of ketone bodies may be beneficial for mothers and children. One of the possible biological benefits of morning sickness is the prevention of diseases that have an increased incidence due to weight gain during pregnancy.

## 1. Introduction

Approximately 80% of pregnant women develop morning sickness, with 0.5–0.8% becoming severe enough to necessitate hospitalization [1,2,3]. The incidence of severe morning sickness in East Asian women is approximately 3.6% [4], which is slightly higher than the incidence range (0.3–2.0%) in other races [4]. Various endocrine substances, including progesterone and human chorionic gonadotropin (HCG), have been investigated as causative agents of morning sickness; however, no definitive causative agent has been identified yet [5]. Physiological changes during pregnancy have some advantages for the mother and child: for example, anemia during pregnancy is the result of increased circulating blood volume and facilitates the prevention of blood clots. The higher the circulating plasma volume, the less likely gestational hypertension or intrauterine growth restriction will occur [6]. However, the physiological benefits of morning sickness are unknown, and severe cases can even be fatal for the mother.

In patients with severe morning sickness, ketone bodies are detected on urinalysis. Ketone bodies can pass through the placenta and become a source of nutrition for the fetus [7]. In adults, ketones have been shown to be beneficial for the central nervous system and can affect many metabolic processes associated with aging and apoptosis. One of the ketone bodies, 3-hydroxybutyric acid, blocks NLR family pyrin domain containing 3 (NLRP3) inflammasome and attenuates caspase-1 and interleukin-1β secretion in mouse models and is believed to reduce hypoglycemia-related neuronal apoptosis, increase the number of motor neurons, increase neuronal activity and angiogenesis, and protect neuronal cell cultures from the development of amyloid pathology [8,9,10]. Ketone bodies pass through the placenta and are consumed by the brain, and thereby contribute to the growth of the fetus’ central nervous system [11]. We focused on the hypothesis that morning sickness begins at the same time as when the maternal serum ketone levels increase and neural tube formation occurs in the fetus. It has been reported that cell movement of the surrounding tissue (non-neural ectoderm), which does not become a neural tube, is also essential for neural tube formation [12], and it was thought that the effect of ketone bodies on neural tube formation also affects tissues other than the neural tube and may affect the fetal body shape.

The HCG level increases in early stage pregnancy, and this increase is considered one of the causes of morning sickness [13]—a theory that is supported by the fact that the highest incidence of morning sickness is observed at peak HCG levels, as well as in twin pregnancies and hydatidiform moles, which induce high HCG levels [14]. The incidence of hyperemesis gravidarum in singleton pregnancies is reported as 1.4%, while that in twins is 2.7%, which is about twice as high [14]. Therefore, the current data in the literature suggests a relationship between morning sickness and high HCG levels; however, the role HCG plays in the pathogenesis of morning sickness remains unclear [5].

We hypothesized that an increase in the levels of maternal serum ketone bodies and HCG in early pregnancy is possibly beneficial for the fetus. Therefore, this study was conducted to investigate the biological benefits of morning sickness by comparing the levels of serum ketone bodies and HCG during early pregnancy with factors related to neonatal growth.

## 2. Materials and Methods

This study was approved by the Institutional Review Board of the Hokkaido University Hospital (019-0390). All participants provided written informed consent prior to their participation in the study. The study enrolled 379 pregnant women who were scheduled to give birth in Obihiro-Kosei General Hospital, Hakodate Central General Hospital, or the Japan Community Health Care Organization Hokkaido Hospital between October 2018 and April 2019. Physicians at the study sites selected participants based on the following inclusion criteria: (1) those aged 18 years or above, (2) those scheduled to give birth at participating institutions, and (3) those who agreed to participate in the study. The exclusion criteria were as follows: (1) those with complications from earlier pregnancy and (2) those with a gestational age of 14 weeks or above.

Blood samples were collected at the time of regular blood tests in the first (8–12 gestational weeks), second (24–27 gestational weeks), and third trimesters (35–37 gestational weeks), and the serum levels of ketone bodies (total ketones, acetoacetic acid, and 3-hydroxybutyric acid) and HCG were quantified. In addition, the participant’s age, height, pre-pregnancy weight, pre-pregnancy body mass index (BMI), weight at blood sampling, weight at delivery, BMI at delivery, pregnancy and delivery history, weeks of delivery, delivery pattern, placental weight, and neonatal findings (e.g., sex, birth weight, birth length, birth head circumference, birth chest circumference) were recorded from the medical records. The relationship of morning sickness with serum levels of ketone bodies and HCG was retrospectively examined.

In the analysis phase, pregnant women who delivered after 36 weeks of gestation were included, whereas patients with twin pregnancies, thyroid disease, preeclampsia, kidney disease, and defect data were excluded.

### 2.1. Biochemical Procedures 

Serum was stored at −80 °C until assays were conducted for the following four blood variables: total ketone bodies, acetoacetic acid, 3-hydroxybutyric acid, and HCG levels were measured using enzyme linked immune sorbent assay kits TKB-L (KAINOS, Tokyo, Japan), 3HB-L (KAINOS, Tokyo, Japan), and II HCG (TOSOH, Tokyo, Japan), respectively.

### 2.2. Statistical Methods

Statistical analyses were performed using the JMP Pro14© statistical software package (SAS, Cary, NC, USA). Changes in variables within a group were compared using the *t*-test and the Tukey–Kramer method. Single regression analysis was used to investigate the relationship between ketone bodies and perinatal prognosis. In all analyses, statistical significance was set at *p* < 0.05. 

## 3. Results

Among the 379 pregnant women who participated in the study, 197, 94, and 88 were screened from Obihiro-Kosei General Hospital, Hakodate Central General Hospital, and the Japan Community Health Care Organization Hokkaido Hospital, respectively. However, after screening, 105 eligible women who missed blood sampling during the first, second, or third trimester (including preterm birth at <36 weeks of gestation), 6 women with twin pregnancies, 15 with thyroid disease, 4 with preeclampsia, 2 with kidneys disease, and 2 with missing data in the medical records were excluded; thus, a total of 245 women, including 11 with gestational diabetes, were included in the final analysis dataset (Figure 1).

### 3.1. Demographic Characteristics

Among the 245 participants, 85 were primiparas. The mean (standard deviation) age and gestational period at delivery were 31.9 ± 5.0 years and 39.1 ± 1.2 gestational weeks, respectively. The timing of blood sampling was 10.2 ± 1.5, 25.8 ± 1.3, and 36.2 ± 0.9 gestational weeks for the first, second, and third trimesters, respectively. There were 13 deliveries at 36 gestational weeks, 191 transvaginal deliveries, and 54 cesarean deliveries (Table 1).

### 3.2. Hematological Parameters during Pregnancy

The median 3-hydroxybutyric acid levels were 27.8, 21.2, and 42.2 μmol /L in the first, second, and third trimesters, respectively. The median concentrations of acetoacetic acid, which reflects the ketone body index for a longer period than the 3-hydroxybutyric acid, were 18.6, 19.9, and 24.1 μmol/L in the first, second, and third trimesters, respectively. The median HCG levels were 132,000, 16,400, and 22,300 IU/L for the first, second, and third trimesters, respectively. The 3-hydroxybutyric acid concentration decreased in the second trimester, but the acetoacetic acid levels showed a tendency to gradually increase from the first to the third trimester. As a 50 g glucose challenge test (GCT) is conducted at the time of regular blood sampling in the second trimester, it was considered that the blood sampling at 60 min after consuming 50 g glucose affected the levels of total ketone bodies and decreased the 3-hydroxybutyric acid level. However, as acetoacetic acid has a long half-life, it is considered to have been less affected by the glucose challenge test (Table 2).

### 3.3. Effect of Glucose Challenge Test on Ketone Body Concentrations

The 50 g GCT in the second trimester was not conducted in the Obihiro-Kosei General Hospital, and the random blood glucose level was used to screen for gestational diabetes. We compared the blood parameters of participants in the second trimester at the Obihiro-Kosei General Hospital (without GCT) and the other participants (with GCT). The median 3-hydroxybutyric acid level was significantly higher among the participants in the without GCT group than among those in the with GCT group (26.0 vs. 16.3 µmol/L, *p* < 0.001) (Table 3). Therefore, when the change in the median 3-hydroxybutyric acid level during pregnancy was confirmed only in the without GCT group, that is, where there was no glucose intake prior to blood sampling, the level gradually increased in the order of the first, second, and third trimester (25.4, 27.1, and 41.7 µmol/L, respectively; Figure 2). 

The 3-hydroxybutyric acid level during pregnancy in the without glucose challenge test group gradually increased in the order of the first, second, and third trimester.

### 3.4. Changes in HCG during Pregnancy

The serum HCG level was high in the first trimester, peaked at 8 weeks of gestation, and decreased gradually, but did not change considerably from the second to the third trimester (Figure 3 and Appendix A).

The serum HCG level was high in the first trimester, peaked at 8 weeks of gestation, and decreased gradually

### 3.5. Relationship between Ketone Body and HCG in Early Pregnancy and the Neonatal Body Shape

We examined the correlations between the neonatal body shape and the serum levels of 3-hydroxybutyric acid, acetoacetic acid, and HCG in the first trimester, but found no significant correlation with the birth weight, birth length, birth head circumference, or birth chest circumference (Figure 4).

There were no correlations between the neonatal body shape and the serum levels of 3-hydroxybutyric acid, acetoacetic acid, and HCG in the first trimester.

### 3.6. Relationship between Placental Weight and Other Factors

No correlation was found between placental weight and the serum levels of 3-hydroxybutyric acid or HCG. There was a positive correlation between the placental weight and birth weight (Figure 5).

### 3.7. Gestational Characteristics of the Weight-Loss and Non-Weight-Loss Groups

For an evaluation of the severity of morning sickness, we compared whether there was any weight loss during pregnancy: 88 women lost weight, whereas 157 did not. In the weight-loss group, the levels of total ketone bodies, 3-hydroxybutyric acid, and acetoacetic acid in the first trimester were significantly higher than the levels in the non-weight-loss group; however, there was no significant intergroup difference in the perinatal prognosis. Although the weight loss group had a significantly high pre-pregnancy BMI, there was no significant intergroup difference in the BMI at delivery (Table 4).

## 4. Discussion

This study showed that: (1) serum levels of ketone bodies gradually increased during pregnancy; (2) the serum levels of ketone bodies and HCG in the first trimester did not affect the neonatal body shape; and (3) women who lose weight in early pregnancy have a higher pre-pregnancy BMI, which may lead to physiological benefits from morning sickness that could prevent complications.

To our knowledge, this is the first report to show that the serum levels of ketone bodies gradually increased during pregnancy. The abovementioned trend was particularly evident in institutions that did not perform the GCT during the second trimester. There was a significant difference in the serum levels of ketone bodies between institutions that performed the GCT in the second trimester and those that did not, suggesting that sugar intake affected the results of the blood tests (Table 3). Ketone bodies pass through the placenta and are consumed by the brain, thereby contributing to the growth of the fetal central nervous system [11]. Furthermore, ketone bodies act protectively on brain cells in Alzheimer’s and Parkinson’s diseases [15]. Moreover, a ketogenic diet may reduce the frequency of epileptic seizures [16,17]. In contrast, a study showed that an elevated 3-hydroxybutyric acid level in the third trimester adversely influenced the intelligence of the child [18]. In our study, an increase in the serum levels of ketone bodies in the third trimester of pregnancy was observed as a physiological change, which suggested that a high level of ketone bodies is unlikely to adversely affect the fetus.

In this study, we examined the hypothesis that ketone bodies, which are by-products of morning sickness, and, HCG, which is a possible cause of morning sickness, may confer benefits on fetal development. However, there was no correlation between the serum levels of ketone bodies or HCG in the first trimester of pregnancy, when morning sickness is most severe, and the body shape, as ascertained by the birth weight, length, head circumference, and chest circumference of the newborn (Figure 4). We inferred that the higher the level of ketone bodies in early pregnancy, the larger is the fetal brain. By focusing on the neuroprotective effects of ketone bodies [14,15,16], we predicted that a higher ketone body concentration in early pregnancy, the greater would be the head circumference of the neonate; however, we found no such relationship. Nonetheless, birth head circumference only may be insufficient as an indicator of brain growth, but it is difficult to assess brain function at this timepoint. HCG is produced by the villous synctiotrophoblast cell layer [19] and, therefore, we expected a correlation between the HCG concentration and the placental weight, but there was no significant relationship. There was a positive correlation between placental weight and birth weight, which was previously reported [20,21] (Figure 5). HCG levels peaked at 8 weeks of gestation, which is consistent with reports from international studies [22] and shows that the trend of HCG levels is the same in Japanese women as in Western women.

We focused on weight loss during pregnancy as a method for evaluating the severity of morning sickness. On comparing the group that had weight loss and the group that did not, we found that the group with weight loss had significantly higher serum 3-hydroxybutyric acid concentrations in the first trimester. The pre-pregnancy BMI was slightly higher in the group with weight loss during pregnancy than in the non-weight-loss group (Table 4). This result was consistent with previous studies, which showed that a higher pre-pregnancy BMI increases the risk of severe morning sickness [23]. However, this finding was contradicted by a study that only included pregnant women in East Asia and found that women with a lower pre-pregnancy BMI were more likely to experience hyperemesis gravidarum [4]. Considering the biological benefits, it may be logical for pregnant women with a high pre-pregnancy BMI to lose weight during pregnancy. The higher the pre-pregnancy BMI, the greater the risk of gestational diabetes and preeclampsia [24,25]. Furthermore, it is well known that greater weight gain during pregnancy is associated with a higher likelihood of developing gestational diabetes and preeclampsia [24,26,27]. One of the benefits of morning sickness may be to reduce weight gain during pregnancy. In future studies, it may be necessary to distinguish whether there are physiological reasons for the weight loss mediated by morning sickness.

There are several limitations of this study. First, weight loss during pregnancy was used as an evaluation method for morning sickness. However, weight loss information is not an adequate indicator of morning sickness. Second, we also excluded cases of preterm delivery, preeclampsia, and several diseases; therefore, the relationship between initial ketone body levels and these diseases remains unclear. Third, a research design to explore the function aspect would be desirable, but it would require monitoring of the growth of newborns and changes in neurological development, which was difficult considering the short study period. Since formation of the neural tube requires cell movement in the surrounding tissues that do not form the neural tube [12], we thought that ketone bodies affect the formation of not only the neural tube but also the whole fetus, and thus we performed this study. However, assessment of neonatal morphology is insufficient as a substitute for assessment of function, and further studies with sufficient time are needed. Of note, it is possible to infer the effects of ketone bodies during pregnancy on the development of the child’s brain, and further research is needed to clarify the relationship.

## 5. Conclusions

During pregnancy, serum levels of ketone bodies increased from early to late pregnancy and peaked in late pregnancy. As the elevation of serum levels of ketone bodies in the third trimester is a physiological change, high serum levels of ketone bodies may be beneficial for both the mother and the child. However, serum levels of ketone bodies and HCG in early pregnancy had no effect on the neonatal body shape. One of the biological benefits of morning sickness in early pregnancy was the prevention of diseases that possibly increase in frequency due to excessive weight gain during pregnancy. This study did not examine the effects of maternal ketone bodies on fetal neurodevelopment, and further studies are warranted.

## Figures and Tables

**Figure 1 nutrients-14-01971-f001:**
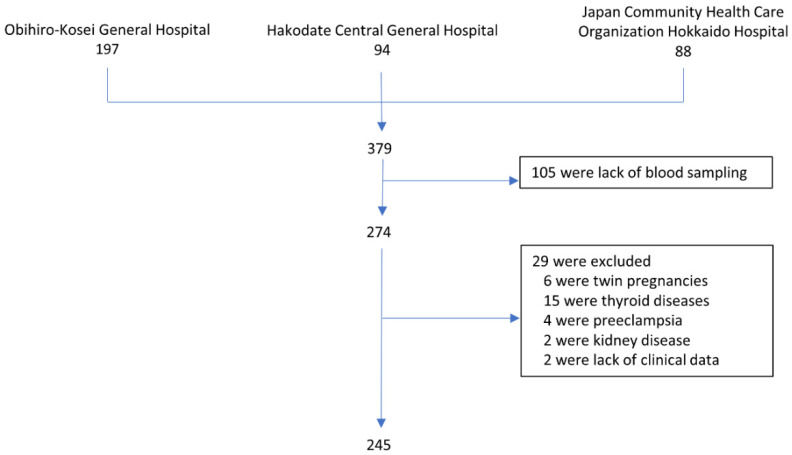
Study design.

**Figure 2 nutrients-14-01971-f002:**
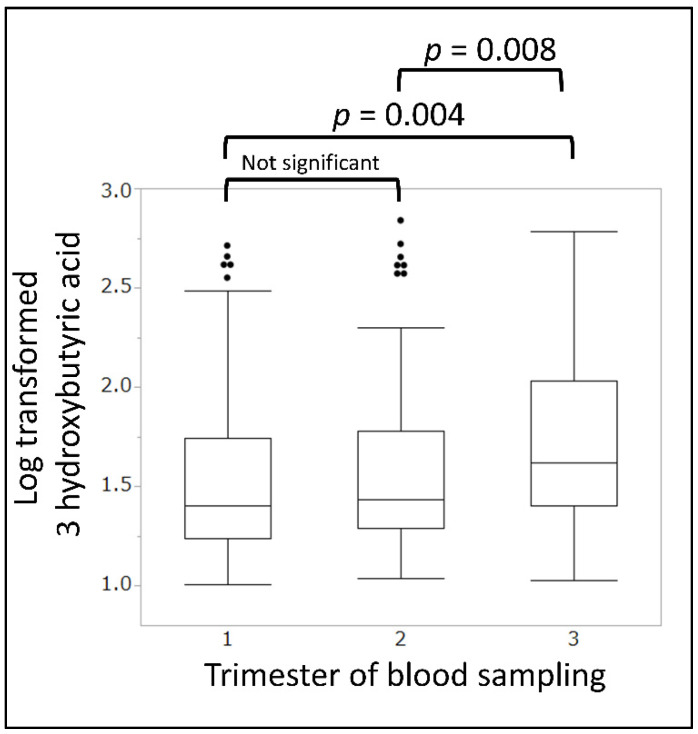
Median 3-hydroxybutyric acid levels in the group without the glucose challenge test during pregnancy.

**Figure 3 nutrients-14-01971-f003:**
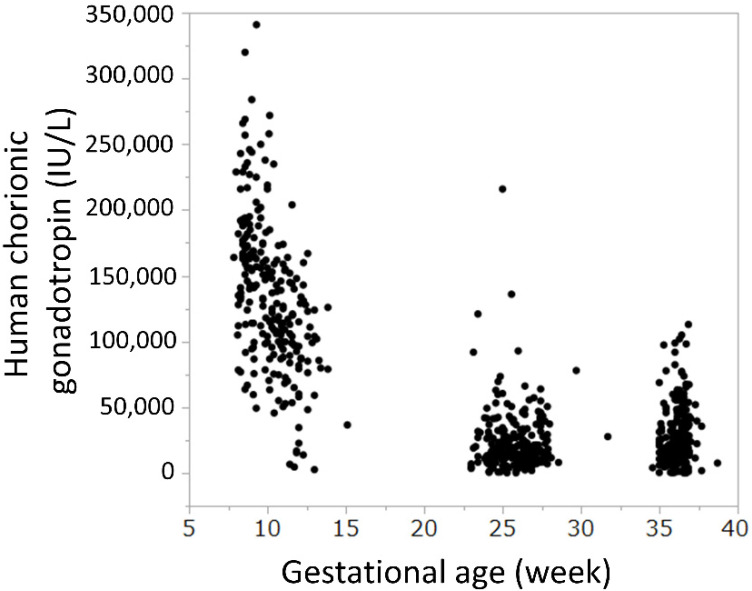
Serum human chorionic gonadotropin levels during pregnancy.

**Figure 4 nutrients-14-01971-f004:**
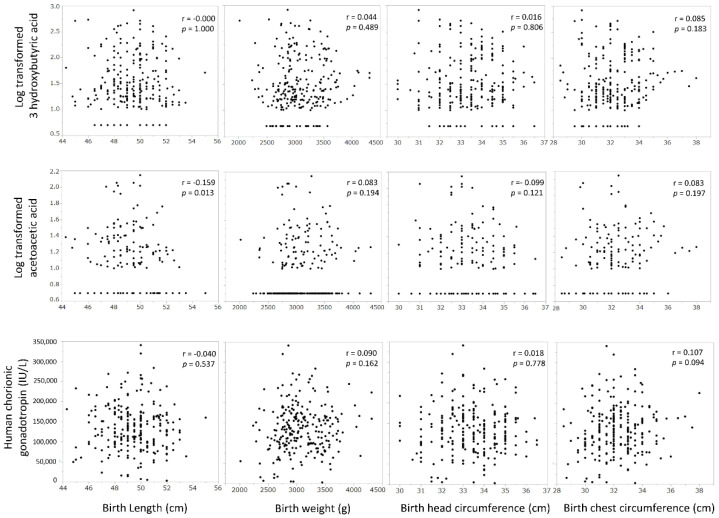
Relationship between serum levels of ketone bodies or human chorionic gonadotropin in the first trimester with the neonatal body shape.

**Figure 5 nutrients-14-01971-f005:**
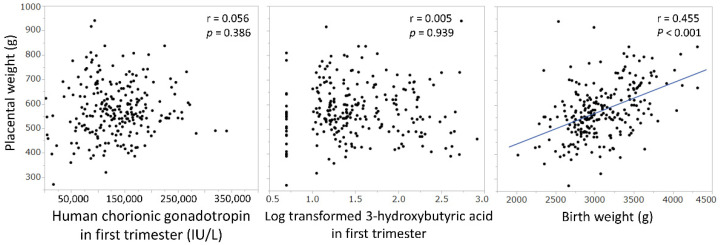
Correlation of placental weight with serum levels of human chorionic gonadotropin and 3-hydroxybutyric acid, and birth weight.

**Table 1 nutrients-14-01971-t001:** Baseline characteristics of 245 pregnant women.

Nulliparous women	85, 35%
Age, years	31.9 (5.0)
Height, m	1.58 (0.05)
Pre-pregnancy weight, kg	53.1 (10.3)
Pre-pregnancy body mass index, kg/m^2^	21.3 (3.5)
Weight gain in pregnancy, kg	11.2 (3.5)
Gestational week at delivery, week	39.1 (1.2)
Preterm delivery at 36 gestational weeks	13 (5.3%)
Vaginal delivery	191 (78%)
Cesarean delivery	54 (22%)
Infant sex	
Male	125, 51%
Birth weight, kg	3.1 (0.4)
Birth length, cm	49.4 (1.8)
Birth head circumference, cm	33.4 (1.3)
Birth chest circumference, cm	32.2 (1.6)
Timing of the tests	
First trimester, week	10.2 (1.5)
Second trimester, week	25.8 (1.3)
Third trimester, week	36.2 (0.9)

Data are presented as the means (standard deviation).

**Table 2 nutrients-14-01971-t002:** Hematological data of 245 pregnant women.

	1st Trimester	2nd Trimester	3rd Trimester	*p*-Value
**Clinical data**				
Maternal body weight, kg	53.8 (10.4)	48.6 (9.8)	62.6 (10.1)	<0.001
Weight gain during pregnancy, kg	0.7 (2.4)	5.6 (3.2)	9.5 (3.9)	<0.001
Weight gain, %	1.4 (4.5)	11.1 (6.6)	18.8 (8.4)	<0.001
Body mass index, kg/m^2^	21.6 (3.6)	23.5 (3.3)	25.0 (3.4)	<0.001
Gestational age, week	10.2 (1.5)	25.8 (1.3)	36.2 (0.9)	<0.001
**Hematological data**				
Total ketones, µmol/L	39.3 (5.0–934)	32.8 (5.0–812)	58.8 (5.0–1460)	<0.001
3-hydroxybutyric acid, µmol/L	27.8 (5.0–821)	21.2 (5.0–690)	42.2 (5.0–1420)	<0.001
Acetoacetic acid, µmol/L	18.6 (5.0–140)	19.9 (5.0–146)	24.1 (5.0–161)	0.083
Human chorionic gonadotropin, IU/L	132,000 (2620–341,000)	16,400 (166–216,000)	22,300 (31.9–113,000)	<0.001

Clinical data are presented as mean (standard deviation). Hematological data are presented as median (range). Data below the measured value were set to 5.

**Table 3 nutrients-14-01971-t003:** Comparison of hematological parameters in the second trimester in the groups with and without a glucose challenge test.

	without GCT	with GCT	*p*-Value
**Clinical Data**			
Number of women	105	140	
Age, years	31.8 (4.7)	32.2 (5.2)	0.196
Height, m	157.2 (0.5)	158.3 (0.5)	0.115
Pre-pregnancy weight, kg	52.3 (9.8)	53.7 (10.8)	0.277
Body weight in the second trimester, kg	58.5 (9.5)	58.9 (10.2)	0.732
Body mass index, kg/m^2^	23.6 (3.1)	23.5 (3.5)	0.758
Weight gain in the second trimester, kg	6.1 (3.2)	5.1 (3.1)	0.014
Weight gain, %	12.3 (6.9)	10.2 (6.3)	0.012
Gestational age, weeks	26.8 (0.6)	25.0 (1.1)	<0.001
**Hematological data**			
Total ketones, µmol/L	46.7 (12.3–812)	18.9 (5–296)	<0.001
3-hydroxybutyric acid, µmol/L	26.0 (5–690)	16.3 (5–271)	<0.001
Acetoacetic acid, µmol/L	19.3 (5–146)	5.0 (5–25.3)	<0.001
Human chorionic gonadotropin, IU/L	16,300 (1890–92,900)	16,550 (166–216,000)	0.191

Clinical data are presented as mean (standard deviation). Hematological data are presented as median (range). Data below the measured value were set to 5. GCT, glucose challenge test.

**Table 4 nutrients-14-01971-t004:** Characteristics of the weight-loss and non-weight-loss groups during pregnancy.

	Weight Loss during Pregnancy	Non-Weight Loss	*p*-Value
**Clinical data**			
Number of women	88	157	
Height, m	1.58 (0.05)	1.58 (0.06)	0.345
Pre-pregnancy body weight, kg	54.8 (11.8)	52.2 (9.3)	0.058
Pre-pregnancy body mass index, kg/m^2^	22.0 (4.2)	20.9 (3.1)	0.025
Minimum body weight, kg	53.0 (11.4)	52.2 (9.3)	0.537
Weight loss, kg	1.78 (2.0)	not available.	
Weight gain during pregnancy, %	9.8 (2.9)	12.1 (3.6)	<0.001
Body weight at delivery, kg	62.8 (11.1)	64.3 (9.8)	0.274
Body mass index at delivery, kg/m^2^	25.2 (3.8)	25.8 (3.2)	0.193
Gestational age at delivery, week	39.2 (1.1)	39.0 (1.2)	0.362
Birth length, cm	49.7 (1.6)	49.3 (1.9)	0.091
Birth weight, kg	3.08 (0.35)	3.08 (0.41)	0.981
Birth head circumstance, cm	33.6 (1.3)	33.3 (1.3)	0.114
Birth chest circumstance, cm	32.3 (1.5)	32.2 (1.7)	0.397
**Hematological data at first trimester**			
Total ketones, µmol/L	46.0 (5–934)	34.0 (5–541)	<0.001
3-hydroxybutyric acid, µmol/L	33.0 (5–821)	23.4 (5–539)	<0.001
Acetoacetic acid, µmol/L	5.0 (5–140)	5 (5–113)	0.046
Human chorionic gonadotropin, IU/L	137,000 (4640–341,000)	129,000 (2620–284,000)	0.072

Clinical data are presented as mean (standard deviation). Hematological data are presented as median (range). Data below the measured value were set to 5.

## Data Availability

The data presented in this study are available on request from the corresponding author. The data are not publicly available due to privacy.

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
