# Peer review of "Changes in Serum Levels of Ketone Bodies and Human Chorionic Gonadotropin during Pregnancy in Relation to the Neonatal Body Shape: A Retrospective Analysis"

_nutrients, 2022, doi:10.3390/nu14091971_

Round 1

Reviewer 1 Report

I congratulate the authors on this very interesting paper.

This field of research is very innovative.

here are provided with my personal suggestion and comments:

1-Lines 48-52 add that the effects named are present in murine models.

2-lines 86-90 please better explain what are TKB-L,3HB-L and II HCG in the biochemical procedure section.

3-Table 4, delete or insert in supplementary materials the rows with only one test (i.g. week 7, 15 etc...), these data do not have a clinical significance.

4-line 258 a typo in the word 3-hydroxybutyric acid is present.

Otherwise, the introduction must be improved with more references.

Author Response

  • Lines 48-52 add that the effects named are present in murine models.

Response: Thank you for your comment. We have revised that phrase. We have also added a description of the NLRP3 inflammasome.

  • lines 86-90 please better explain what are TKB-L,3HB-L and II HCG in the biochemical procedure section.

Response: Thank you for your comment. We have revised the sentence, as follows: “total ketone bodies, acetoacetic acid, 3-hydroxybutyric acid, and HCG levels were measured using enzyme linked immunosolvent assay kits TKB-L (KAINOS, Tokyo, Japan), 3HB-L (KAINOS, Tokyo, Japan), and â…¡ HCG (TOSOH, Tokyo, Japan), respectively.”

  • Table 4, delete or insert in supplementary materials the rows with only one test (i.g. week 7, 15 etc...), these data do not have a clinical significance.

Response: Thank you for your comment. Table 4 has been changed to Supplementary Table 1.

  • line 258 a typo in the word 3-hydroxybutyric acid is present.

Response: Thank you for your comment. We have revised the mentioned word.

Otherwise, the introduction must be improved with more references.

Response: Thank you for your comment. We have added some sentences and references.

Reviewer 2 Report

The manuscript "Changes in Serum Levels of Ketone Bodies and Human 
Chorionic Gonadotropin During Pregnancy in Relation to the 
Neonatal Body Shape: A Retrospective Analysis" is an interesting manuscript on  the relationship between serum levels of ketone 18
bodies and HCG in the first, second, and third trimesters and neonatal body shape.

The field of the review is extremely interesting and it has a great interest to a general audience, but it looks confusing in some points of the text, so they need to be clarified.

 The authors declare that "they focused on the hypothesis that morning sickness begins at the same time as that the maternal serum ketone levels increase and neural tube formation occurs in the fetus": it is ambitious but in the results of the study they have no elements to confirm their hypothesis, because the authors focus mainly on the head circumference of the neonate, so basically on a growth aspect, not a functional one.

What does it mean" time of regular blood tests in the first, second, and third trimesters"? Which weeks of gestation do the authors consider? Particularly in the first trimester, considering that there is high variability in the beta hcg values between the different weeks, it is important to specify the exact time of the blood samples.

The inclusion and exclusion criteria are not well explained; why, for example, gestational diabetes was not considered? It is mandatory to describe clearly the inclusion and exclusion criteria, for patients and pregnancies pathologies.

The authors have not adequately highlighted the strengths and limitations of their study. I suggest better specifying these points in the discussion and the conclusion of this work. 

Author Response

 The authors declare that "they focused on the hypothesis that morning sickness begins at the same time as that the maternal serum ketone levels increase and neural tube formation occurs in the fetus": it is ambitious but in the results of the study they have no elements to confirm their hypothesis, because the authors focus mainly on the head circumference of the neonate, so basically on a growth aspect, not a functional one.

Response: Thank you for your comment. A research design to explore the function aspect would be desirable, but it would require monitoring of the growth of newborns and changes in IQ, which was difficult considering the short study period. Since formation of the neural tube requires cell movement in the surrounding tissues that do not form the neural tube, we thought that ketone bodies affect the formation of not only the neural tube but also the whole fetus. We designed this study to compare the body shape of newborns with serum levels of ketone bodies. We have added the following sentence in the Introduction: “It has been reported that cell movement of the surrounding tissue (non-neural ectoderm), which does not become a neural tube, is also essential for neural tube formation [Morita H, et al.], and it was thought that the effect of ketone bodies on neural tube formation also affects tissues other than the neural tube and may affect the fetal body shape.”

What does it mean" time of regular blood tests in the first, second, and third trimesters"? Which weeks of gestation do the authors consider? Particularly in the first trimester, considering that there is high variability in the beta hcg values between the different weeks, it is important to specify the exact time of the blood samples.

Response: We are sorry for the unclear description. We have revised the phrase as follows: “Blood samples were collected at the time of regular blood tests in the first (8–12 gestational weeks), second (24–gestational weeks), and third trimesters (35–37 gestational weeks),”. The timing of the tests is also described in Table 1 and Supplementary table 1.

The inclusion and exclusion criteria are not well explained; why, for example, gestational diabetes was not considered? It is mandatory to describe clearly the inclusion and exclusion criteria, for patients and pregnancies pathologies.

Response: Thank you for your comment. We have added more details about the exclusion and inclusion criteria in the Material and Methods: Physicians at the study sites selected participants based on the following inclusion criteria: (1) those aged 18 years or above, (2) those scheduled to give birth at participating institutions, and (3) those who agreed to participate in the study. The exclusion criteria were as follows: (1) those with complications from earlier pregnancy and (2) those with a gestational age of 14 weeks or above.” We have also added the following details in the Results: “However, after screening, 105 eligible women who missed blood sampling during the first, second, or third trimester (including preterm birth at <36 weeks of gestation), 6 women with twin pregnancies, 15 with thyroid disease, 4 with preeclampsia, 2 with kidneys disease, and 2 with missing data in the medical records were excluded; thus, a total of 245 women, including 11 with gestational diabetes, were included in the final analysis dataset (Figure 1).

The authors have not adequately highlighted the strengths and limitations of their study. I suggest better specifying these points in the discussion and the conclusion of this work.

Response: Thank you for your comment. We have added these details in the Discussion and Conclusions.
